# Interaction Between α-Synuclein and DJ-1 in Parkinson’s Disease

**DOI:** 10.3390/brainsci15090899

**Published:** 2025-08-22

**Authors:** Pouya Sobhifar, David R. Brown

**Affiliations:** Department of Life Sciences, University of Bath, Bath BA2 7AY, UK; pouyasrt@yahoo.com

**Keywords:** α-synuclein, DJ-1, Parkinson’s disease, aggregation, glycation

## Abstract

Parkinson’s disease (PD) is one of the most common neurodegenerative disorders among the elderly. The exact etiology of sporadic PD is still unknown; however, there is general consensus that the accumulation and aggregation of α-synuclein (α-syn) are among the prominent pathological features. The precise function of α-syn in the healthy human brain is not agreed upon, although it has been reported to play a role in vesicular trafficking and neurotransmitter release. Dutch Juvenile-1 (DJ-1) is a multifunctional protein involved in regulating an array of mechanisms, including oxidative stress, ferroptosis, mitochondrial and dopamine homeostasis. Loss-of-function of DJ-1 was reported to cause familial PD, and oxidative inactivation of DJ-1 has been observed in sporadic cases, suggesting that both genetic and post-translational events converge on common disease pathways. This review proposes that loss of DJ-1 function may elevate intracellular α-syn levels, leading to their aggregation and consequent neurotoxicity. Reports suggest that DJ-1 can inhibit α-syn aggregation, facilitate α-syn clearance via chaperone-mediated autophagy, and act as a deglycase or glyoxalase to neutralize glycated α-syn species. Clinical studies have also reported altered DJ-1 oxidation states in PD patient samples, supporting its potential as a biomarker. By bridging familial and sporadic PD mechanisms, DJ-1 emerges as a compelling therapeutic target with the potential to mitigate α-syn–mediated neurodegeneration across both forms. However, further research is required to fully establish its clinical relevance and translational potential.

## 1. Parkinson’s Disease

PD was initially described by James Parkinson in his “Essay on the shaking palsy” in 1817. After Alzheimer’s disease, PD is the most common neurodegenerative disorder, affecting approximately 2% of the population over 60 years old around the world [1]. Regrettably, PD has no cure to this date, with existing medications such as Levodopa, dopamine agonists, mono-amide oxidase B, and catechol-O-methyltransferase inhibitors only alleviating the symptoms rather than modifying the disease progression [2]. Levodopa is deemed the first-line medication. It is typically combined with Carbidopa, which inhibits its conversion to dopamine peripherally and also reduces the risk of nausea [3]. As of yet, there are no novel medications that result in prevention or stopping PD progression. This is due largely to the inability of current drugs to pass the blood–brain barrier. Nanotechnology seems promising, as nanoparticles are able to pass through the BBB because of their nanosize [4]. Hence nanoparticles can be the future of delivering antiparkinsonism drugs to PD patients.

Clinically, PD manifests through a mixture of motor and non-motor symptoms. Non-motor symptoms, normally emerging in the early stages, include constipation, insomnia, depression, rapid eye movement behavior disorder, and anosmia. In the later stages, motor symptoms become predominant, encompassing bradykinesia, rest tremor, muscle rigidity, and postural impairment [5]. Beyond conventional pharmacological treatments, novel therapeutic strategies are under investigation. For instance, low-frequency whole-body electromyostimulation has been shown to improve physical performance, reduce fatigue, and decrease serum α-synuclein while increasing BDNF levels in PD patients, offering a potential therapeutic approach to mitigate both motor and non-motor symptoms [6].

Neuropathologically, PD has two main features involving the presence of intraneuronal inclusions known as Lewy bodies, primarily containing insoluble aggregated forms of α-syn, and the degeneration of dopaminergic neurons in the substantia nigra (SN) pars compacta of the midbrain [3]. There are two types of PD: sporadic and familial PD. While 80–90% of all PD cases are sporadic, only a minority (10–20%) of them can be attributed to genetic mutations. Although the exact etiology of sporadic PD is still unidentified, it is believed to be due to a complex interplay of environmental and genetic factors. Environmental factors may simply be an exposure to neurotoxins, pesticides, and herbicides, such as rotenone and paraquat [2].

To date, numerous PD-causative genes have been identified, among which six are most frequently cited: α-syn (PARK1), Parkin (PARK2), PTEN-induced putative kinase 1 (PINK1), DJ-1 (PARK7), Leucine-rich repeat kinase 2 (LRRK2), and Glucocerebrosidase (GBA) [1].

Despite being initially described over 200 years ago, the cellular and molecular mechanisms responsible for PD remain inadequately understood. It was reported in 2003 by Bonifati and colleagues that the absence and mutation of DJ-1 caused an early onset of familial PD in humans. Knowing that α-syn plays a key role in PD pathogenesis, this review hypothesizes that DJ-1 is essential for preventing the formation of pathological α-syn. Therefore, loss-of-function of DJ-1, whether due to genetic mutations or oxidative inactivation, may lead to α-syn aggregation and eventually PD development.

## 2. α-Synuclein

α-syn is encoded by the *SNCA* gene produced in the human brain. It comprises approximately 0.5–1% of the total amount of cytosolic proteins in neurons [7]. It is primarily expressed at presynaptic nerve terminals. α-syn can also be sparsely detected in the neuronal cytoplasm and nucleus. It belongs to the synuclein family, consisting of synoretin, α-, β-, and γ-synuclein [8].

Point mutations in *SNCA* (PARK1) and gene multiplication (PARK4) cause autosomal dominant PD. The identified mutation, A53T, was first discovered in four families with early-onset PD and autosomal dominant inheritance [9]. Followed by other point mutations such as A30P, E46K, H50Q, G51D, A18T, and A29S. These mutations occur very rarely [10].

Compared with point mutations, *SNCA* multiplications are slightly more prevalent. In 2003, Singleton and colleagues, for the first time, identified a family with *SNCA* triplication causing early-onset PD. Finally, the *SNCA* gene duplication was reported to cause late-onset PD [11,12].

### 2.1. Structure

α-syn has a molecular weight of 14.5 kDa and is a small 140-amino acid-residue protein. α-syn contains three domains: an N-terminal (1–60 amino acid residues), a hydrophobic central region (61–95 residues), and a C-terminal (96–140 residues) (Figure 1). The N-terminal has seven highly conserved KTKEGV repeats forming an amphipathic α-helix upon membrane binding. Notably, all of the PD-linked mutations mentioned in the previous section are located within the N-terminal domain. These mutations destabilize the protein and increase its susceptibility to aggregation by promoting β-sheet formation, altering secretion, and enhancing fibril propagation [8]. In contrast to the N-terminal, the C-terminal and the central region are less conserved among synuclein family members in mammals. The hydrophobic central region, also known as the non-amyloid-β component, enables β-sheet formation and represents the most aggregation-prone segment of the protein [8].

The C-terminal of the domain is rich in acidic residues, making it hydrophilic in nature. The C-terminal has manifold roles, including regulating interactions with chaperone proteins, controlling aggregation, and modulating fibrilization [13,14]. Following the emergence of evidence of truncated forms of α-syn in the normal and PD patients’ brains [15,16], Zhang et al. (2022) recently examined a C-terminally truncated form of α-syn (1–99) in vitro and found it to be more prone to aggregation, fibrilization, and binding to negatively charged mitochondrial membranes via its N-terminal [13]. Previously, many studies stated that protein disulfide isomerase (PDI) can act as a chaperone to prevent α-syn aggregation via binding to the N-terminal of full-length α-syn [17,18]. Zhang et al. (2022) showed that PDI is less efficient in inhibiting the truncated form of α-syn aggregation. PDI binds to two different segments in the N-terminal. These segments are residues 1–20 and residues 36–42 of the α-syn protein. Association with the first segment promotes α-syn fibrilization, whereas the association with the second segment inhibits it [13]. This indicates that PDI preferentially interacts with the first segment in shorter α-synuclein forms, thereby facilitating α-syn fibrilization. A less truncated form of α-syn (1–119) was shown to alleviate loss of striatal dopamine production (seen with the shorter protein) without affecting dopaminergic neuron viability in transgenic mouse models. This suggests that the shorter variants can cause PD-like symptoms [19]. Aggregates of full-length and truncated α-syn are morphologically very similar, forming straight and unbranched fibrils. However, fibrils from the truncated species are more compact and shorter in length [20].

Recombinant α-syn was initially described in aqueous solutions under physiological conditions as a ‘natively unfolded intrinsically disordered’ monomer acquiring α-helical structure only upon binding to negatively charged membranes [21,22]. However, in 2011, isolated α-syn under non-denaturing conditions from human erythrocytes, mouse brain, and neuronal and non-neuronal cells was illustrated to predominantly exist as helically folded stable tetramers (~58 kDa). Unlike the monomers, the stable tetramers were highly resistant to aggregation [23]. Later, Wang et al. (2011) confirmed these findings, stating that an engineered human α-syn with a 10-residue longer N-terminus formed a dynamic and stable tetramer in the absence of phospholipid bilayer membranes and was resistant to aggregation [24]. In contrast, after analyzing endogenous α-syn from mouse, rat, and human brains and recombinant α-syn produced in *E. coli*. Fauvet et al. (2012) reported that the native α-syn is monomeric [25]. Unlike Bartels et al. (2011) [23] and Wang et al. (2011) [24], Fauvet et al. (2012) [25] employed denaturing conditions and ionic detergents such as SDS, which can disrupt the weak non-covalent interactions between α-syn molecules prior to cell lysis. They also did not apply cross-linkers to viable cells, which could have preserved native oligomeric states during extraction. In 2014, Burro and colleagues demonstrated that α-syn is present as monomers in the cytosol but can undergo a conformational change and multimerize into tetramers and higher-order multimers upon binding to a lipid membrane [26].

**Figure 1 brainsci-15-00899-f001:**
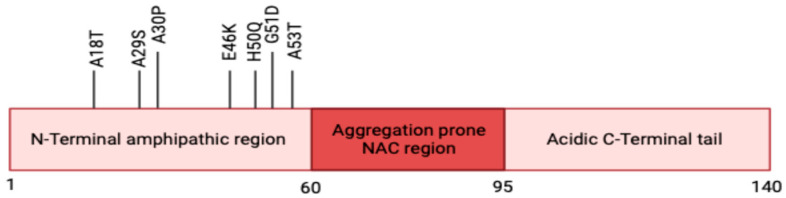
**Schematic representation of α-syn.** α-syn has three domains: an N-terminal, a non-amyloid-β component (NAC), and a C-terminal tail. Mutations linked to familial forms of PD are also depicted (A18T, A29S, A30P, E46K, H50Q, G51D, and A53T) (Created in Biorender.com version 1.3) [27].

### 2.2. Physiological Roles of α-Syn

The precise function of α-syn in a healthy human brain remains unclear. Evidence suggests that α-syn can display both promotive and inhibitory effects on vesicular trafficking and neurotransmitter release. Using α-syn homozygous knockout mice, Abeliovich et al. (2000) reported that nigrostriatal dopamine release recovered faster from repeated neuronal firing compared to genetically unmodified wild-type (WT) mice. These knockout mice also exhibit a mild decrease in striatal dopamine reserves, consistent with an increased release rate [28]. This in vivo experimentation illustrates that α-syn negatively regulates neurotransmitter release. Similarly, a slight overexpression of the α-syn, at such levels that do not result in deposits or noticeable toxicity, was found to inhibit synaptic vesicle exocytosis in murine hippocampal and midbrain dopamine neurons, as observed through optical imaging [28]. Western blot analysis further revealed that α-syn overexpression downregulated synapsin 2 and complexin 2, proteins that accelerate vesicular trafficking during repetitive neuronal firing [29,30]. Contrary to previously reported inhibitory effects, Burré et al. (2010) showed that α-syn can facilitate neurotransmitter release via promoting the docking and fusion of synaptic vesicles with the presynaptic membrane both in vivo and in vitro. Specifically, they illustrated that the C-terminal of α-syn directly binds to the N-terminal of synaptobrevin 2, a v-SNARE protein, promoting SNARE complex assembly [31]. Building on this, Burré et al. (2014) found that membrane-bound α-syn multimerizes and acts as a chaperone, stabilizing the SNARE complex assembly at the synaptic membrane [26].

Besides regulating vesicular trafficking and neurotransmission, overexpression of WT and A53T mutant α-syn in MN9D dopaminergic cells lowered the activity of tyrosine hydroxylase, which is a rate-limiting enzyme in the dopamine biosynthesis process, through direct interaction. Due to the diminished activity of the enzyme, dopamine generation and release were reduced by four times and eight times in WT and A53T mutant α-syn, respectively [32].

In addition, α-syn appears to possess ferrireductase activity, which was reduced primarily in the striatum of PD patients [33]. In 2011, Davies and colleagues provided the initial evidence that α-syn can reduce iron from its ferric (Fe^3+^) to ferrous (Fe^2+^) state using NADH as an electron donor and Cu (ΙΙ) as a cofactor [34]. Since Fe^2+^ serves as a cofactor for tyrosine hydroxylase, this activity could indirectly enhance dopamine production. Later, McDowall et al. (2017) confirmed that α-syn monomers possess *bona fide* ferrireductase activity, with PD-associated E46K and A53T mutations enhancing this enzymatic activity [35]. In vivo expression of human α-syn in rat dopaminergic neurons increased ferrireductase activity, elevated Fe^2+^ levels, and led to accumulation of the toxic dopamine metabolite DOPAL, contributing to neuronal toxicity [33].

### 2.3. α-Syn in Parkinson’s Disease Pathogenesis

There is broad consensus that the accumulation and aggregation of α-syn is the hallmark pathological feature of neurodegenerative disorders such as PD, dementia with Lewy bodies, and Parkinson’s disease dementia—collectively termed as α-synucleinopathies [36]. This section outlines the various mechanisms through which pathological α-syn contributes to the development and progression of PD.

It remains a point of contention in the literature whether cytosolic α-syn exists predominantly as disordered monomers or stable tetramers under normal physiological conditions, although the prevailing view is that it is monomeric. Ko et al. (2008) reported that human α-syn monomers, unlike tetramers, naturally have a high propensity to aggregate, mainly creating insoluble elongated fibrils [37]. This suggests that any shift in the ratio of tetramers to monomers in the brain can lead to α-syn aggregation. In a recent in vitro study, the familial PD mutants (A30P, A53T, E46K) rapidly aggregated into toxic oligomers, whereas WT α-syn aggregated more slowly, favoring the formation of larger oligomers that subsequently converted into fibrils [38]. Winner et al. (2011) compared familial mutants (A30P and A53T) with engineered mutants (E57K and E35K) in a mouse model. While the familial mutants predominantly formed fibrils, engineered mutants favored oligomer formation. Of note, oligomers were significantly more toxic to dopaminergic neurons in the SN [39]. Currently, soluble oligomers rich in β-sheet structures are widely considered the most neurotoxic species in PD [40].

#### 2.3.1. Membrane Disruption and Ion Dysregulation

α-syn oligomers have been reported to create non-selective pores in the cellular membrane, allowing the influx of Ca^2+^, Fe^2+^ and other cations—a concept called the ‘pore-forming’ hypothesis. Using electron microscopy and analytical ultracentrifugation, α-syn, in particular PD-linked A30P and A53T mutants of α-syn, can form annular or tubular protofibrils. These protofibrils are capable of integrating into lipid bilayers to generate pores, potentially leading to ion dysregulation [41]. In support of this, Tsigelny et al. (2012) reported that annular α-syn oligomers, especially those formed by A53T mutants, perforate membranes, creating pore-like structures [42].

The existence of oligomeric α-syn pores remains disputed. Lorenzen et al. (2014) found using ~30 mer stable oligomers that, unlike the annular ones, these species disrupted membrane integrity through interacting with the lipid bilayer, ultimately inducing ion leakage without forming defined pores. The main difference between these papers lies in the type of oligomers employed [43]. Tsigelny et al. (2012) specifically used annular and on-pathway oligomers to fibrils or protofibrils [42], whereas Lorenzen et al. (2014) utilized stable and off-pathway oligomers [43]. Beyond oligomer structure (stable or annular), membrane disruption also depends upon membrane properties such as lipid charge and packing density. For instance, negatively charged membranes and bilayers with loosely packed lipids are more prone to disruption [44].

#### 2.3.2. Neuroinflammation via Microglial and Astrocytic Activation

Results from in vitro studies and mouse models show that extracellular oligomeric α-syn acts as a damage-associated molecular pattern by directly binding to Toll-like receptors 2 and 4 (TLR 2/4) on microglia and astrocytes. This interaction triggers the release of pro-inflammatory cytokines such as IL-1β and TNF-α, as well as ROS including NO and superoxide (O_2_^−^) radicals [45,46]. Persistent TLR 2/4 activation shifts microglia towards a pro-inflammatory M1 phenotype, reducing debris clearance and promoting further cytokine release. This ultimately leads to the impairment of α-syn clearance and exacerbation of neuroinflammation [47].

#### 2.3.3. Mitochondrial Dysfunction

There is evidence that intracellular oligomeric α-syn can impair mitochondrial function and promote neuroinflammation. FILA-1-positive α-syn oligomers at concentrations as low as 1 nM have been shown to inhibit mitochondrial complex I activity and oxidize ATP synthase. This promotes the opening of permeability transition pores. This pore opening allows the influx of Ca^2+^ triggering ROS production, ultimately cell death [48]. However, some studies failed to observe a direct effect of oligomeric α-syn on mitochondrial function. Instead, other reports suggest that oligomeric α-syn indirectly impairs mitochondria by disrupting intracellular Ca^2+^ homeostasis. This disruption results in mitochondrial calcium overload, permeability transition pores opening, and increased ROS production [49,50].

#### 2.3.4. ER Stress and the Unfolded Protein Response

Postmortem analyses of PD brains have revealed upregulation of ER stress markers such as phosphorylated PERK and IRE1α [51]. Toxic α-syn oligomer accumulation within the ER lumen can trigger the unfolded protein response (UPR) through both direct and indirect mechanisms. Under physiological conditions, the ER-resident chaperone protein called Binding Immunoglobulin Protein (BiP/GRP78) associates with UPR sensors (PERK, IRE1α, and ATF6) to maintain them in an inactive state [52]. In PD brains, BiP dissociates from the UPR sensors, leading to their activation, and concurrently binds to α-syn oligomers in an attempt to restore ER homeostasis. However, this interaction with α-syn oligomers impairs BiP’s ability to properly fold proteins, causing ER stress [52]. In a rat model of PD, BiP overexpression significantly reduced the expression of ER stress marker expression (e.g., ATF6 and ATF4), reversing α-syn-induced neurotoxicity [53].

Injecting high-molecular-weight (>100 kDa) toxic α-syn oligomers directly into murine striatum demonstrated that the oligomers bind to calcium-binding protein 1 (CaBP1). This prevents it from inactivating inositol 1,4,5-trisphosphate receptors (IP_3_R) following Ca^2+^ spikes [54]. Further down this pathway, the loss of inactivation leads to Ca^2+^ leakage from the ER into the cytosol, resulting in ER calcium depletion. α-syn oligomers also impaired Sarco/Endoplasmic Reticulum Ca^2+^-ATPase (SERCA) pumps’ activities in vitro by directly binding to them, further exacerbating Ca^2+^ depletion in the ER [55]. Since ER chaperones such as BiP, Calnexin, and Calreticulin require calcium to properly fold proteins, this depletion of Ca^2+^ disrupts protein folding and stimulates the UPR [56]. Hence, α-syn oligomers can also indirectly activate the UPR and induce ER stress through dysregulated calcium signalling.

α-syn overexpression is able to impair Rab1A function, a small GTPase essential for ER-Golgi vesicle trafficking in a rat PD model. The Rab1A impairment leads to Golgi fragmentation and accumulation of unfolded proteins in the ER lumen, which further activates the UPR [57]. α-syn oligomers at the mitochondria-associated ER membranes (MAM) enhance IP_3_R-mediated Ca^2+^ release, driving mitochondrial calcium overload, elevated ROS generation, and eventually activation of apoptotic pathways [58].

#### 2.3.5. Disruption of Proteostasis Pathways

Toxic α-syn oligomers have been shown to impair both the ubiquitin–proteasome system (UPS) and chaperone-mediated autophagy (CMA). Snyder et al. (2003) reported using in vitro and cell-based assays that both monomeric and oligomeric α-syn selectively bind to the S6′ subunit of the 19S proteasome regulatory particle, impairing ubiquitin–dependent proteasomal activity [59]. Moreover, Lindersson et al. (2004) showed that aggregated α-syn binds to 20S proteasome core subunits, inhibiting chymotrypsin-like activity, thereby affecting ubiquitin-independent degradation [60].

Soluble α-syn monomers are primarily degraded by CMA [61]. CMA dysfunction has been reported in PD patients [62]. Under physiological conditions, heat-shock cognate protein of 70 kDa (Hsc70) binds to monomeric α-syn in the cytosol by identifying the KFERQ-like motif. The Hsc70-α-syn complex is subsequently translocated to the lysosomal membrane, where Lysosome-Associated Membrane Protein type 2A (LAMP-2A) receptors directly interact with α-syn. As a result, α-syn is unfolded and translocated into the lysosomal lumen through the LAMP-2A channel in order for it to be degraded by various proteases such as cathepsin D, B, and L [63]. Alvarez Erviti et al. (2010) examined postmortem PD brains, revealing reduced levels of LAMP-2A and Hsc70 in the SN and amygdala. This finding indicates that α-syn degradation is slower in PD patients, increasing the risk of aggregation [60]. Additionally, post-translational modifications such as nitration (e.g., Y39) and oxidation (e.g., M5) of α-syn monomers have been shown to further inhibit both CMA and UPS activity [64,65,66].

#### 2.3.6. Post-Translational Modifications of α-Synuclein

Miranda et al. (2017) detected glycated α-syn in postmortem PD brains and demonstrated that glycation increases with age in mice [67]. Glycation is a non-enzymatic post-translational modification that simply inactivates proteins. A covalent bond forms between cysteine, arginine, and lysine residues in the amino chains of proteins and reactive carbonyls such as reducing sugars (e.g., glucose) or glyoxals (e.g., methylglyoxal). Methylglyoxal (MGO), created endogenously as a by-product of glycolysis, is a highly reactive glycating agent initially transforming proteins into early glycation products (e.g., Schiff bases). Intermediate products can later form that eventually transition into irreversible advanced glycation end products (AGEs; e.g., MG-H1 and CEL) [68]. AGEs interact with membrane-bound receptors called ‘receptors for advanced glycated end points’ (RAGE), activating NFkB and promoting neuroinflammation. In PD murine models, RAGE inhibition alleviates inflammation and rescues dopaminergic neurons in the SN [69]. MGO reacts with dopamine to generate highly reactive compounds such as dopamine-derived tetrahydroisoquinolines (TIQs), which are neurotoxic and implicated in PD [70].

Given that WT α-syn naturally tends to aggregate, glycation of α-syn promotes this aggregation, producing highly toxic oligomers leading to neuronal cell death in PD mouse models [67]. Morphologically, WT α-syn fibrils have straight elongated shapes [37]; nevertheless, glycated α-syn aggregates are mainly amorphous or globular [67]. Glycation chiefly occurs at lysine residues in the N-terminal domain of α-syn adversely, affecting its membrane-binding capability. Moreover, occupation of the same lysine residues by glycation prevents their ubiquitination, hence clearance of α-syn by the ubiquitin–proteasomal system [67]. All these reports show that α-syn aggregation and glycation have a significant role in PD progression.

α-syn phosphorylation is substantially increased within Lewy body inclusions from PD and dementia with Lewy bodies patients compared to healthy controls. This modification occurs especially at serine 129 (Ser129). While approximately 90% of α-syn in Lewy bodies is phosphorylated, only about 4% of phosphorylated α-syn is detected in healthy brains [71]. This suggests tight regulation of phosphorylation under normal conditions but dysregulation in PD. Ser129-phosphorylation promotes α-syn aggregation and fibril formation in vitro. Notably, phosphorylated α-syn at Ser129 subsequently undergoes mono- and di-ubiquitination, which are the predominant forms detected in Lewy bodies [72].

## 3. DJ-1

PARK7 is a PD-associated gene locus found on chromosome 1p36.23, which encodes a 24 kDa protein [73]. The protein, known as DJ-1, is ubiquitously expressed in tissues, including the brain, kidneys, skeletal muscle, and heart. In the human brain, DJ-1 is highly expressed in reactive astrocytes and at relatively lower levels in neurons. It is abundantly found as a homodimer in the cytoplasm and at lower concentrations in the nucleus and mitochondria [74]. In 2003, Bonifati and colleagues identified PARK7 as a causative gene for early-onset familial PD with recessive inheritance. They reported a missense monogenic mutation of L166P and a large 14-kb deletion in the PARK7 gene in Italian and Dutch families, respectively [75]. Furthermore, M26I, D149A [76], A104T [77], E64D [78], and P158Δ [79] DJ-1 point mutations have been associated with early-onset familial PD. L166P, M26I, and P158Δ were reported to hinder the dimerization process of DJ-1, rendering the protein non-functional [80].

### 3.1. Structure

A monomer of DJ-1 is 189-amino acids in length (Figure 2A). The X-ray crystal structure of the DJ-1 monomer reveals that the protein contains 7 β-strands and 9 α-helices in total, referred to as the DJ-1/PfpI domain. Strands β2-β1-β4-β5-β7 are configured in a parallel fashion, while β6 and β7 are arranged anti-parallel, constituting the core. The five central parallel β-sheet strands are sandwiched by eight α-helices, creating a globular, compact, and stable fold (Figure 2B). DJ-1 contains three cysteine residues at amino acid positions 46, 53, and 106. Cysteine residue at position 106 (C106) is a highly conserved nucleophile elbow pocket between α2 and β3 [81]. C106 is a redox-sensitive residue whose oxidation state determines the active level of DJ-1 [82]. Partial oxidation of C106 residue to its sulfinated form (SO_2_H) fully activates the protein. However, the extensive oxidation/hyperoxidization to its sulfonic form (SO_3_H) leads to the aggregation and inactivation [82]. The inactive sulfonic form of DJ-1 was detected in patients with sporadic PD [83]. Laurent et al. (2025) further reported that dopamine-derived TIQs covalently modify and hyperoxidize DJ-1 at C106, rendering the protein inactive [70]. In addition to oxidation, DJ-1 undergoes other post-translational modifications. C46 and C53 residues can be S-nitrosylated under nitrosative stress [84], while SUMOylation at K130 is crucial for DJ-1 to function under oxidative stress [85].

### 3.2. Physiological Roles of DJ-1

Since 2003, DJ-1 has been extensively studied and reported to be involved in multiple intracellular processes, including oxidative stress regulation, mitochondrial and dopamine homeostasis, and ferroptosis [86,87,88].

Mitochondrial Complex Ι activity in the SN of PD patients is impaired [89]. Perturbations in the complex Ι activity lead to inefficient transfer of electrons, reduced ATP production, excessive ROS generation, and potentially neuronal death. The low ATP generation is particularly detrimental for dopaminergic neurons in the pars compacta, which have relatively longer axons and fire rhythmically at a pacemaker-like rate, thereby requiring more energy [90].

DJ-1 acts as an oxidative sensor in the cytoplasm. Upon exposure to elevated levels of ROS, DJ-1 undergoes activation (sulfinated form), coordinating both nuclear and mitochondrial protective responses (Figure 3) [86,87]. Hayashi et al. (2009) found that quiescent DJ-1 stabilizes mitochondrial complex Ι activity through direct binding to the NDUFA4 and ND1 subunits in human SH-SY5Y neuroblastoma cell lines. This interaction was further enhanced under oxidative stress to minimize ROS production. DJ-1 knockdown also reduced complex Ι activity [86], suggesting DJ-1’s essential role in maintaining mitochondrial function under both physiological and stress conditions.

Nuclear roles for DJ-1 have also been suggested (Figure 3). Wang et al. (2011) reported that DJ-1 in response to oxidation may physically interact with ERK1/2, promoting Elk1-mediated transcription of antioxidant enzyme superoxide dismutase 1 (SOD1). SOD1 protects cells from O_2_^−^ radicals [87]. DJ-1 can also promote the nuclear factor erythroid 2–related factor 2 (Nrf2) activity, which is a master regulator of antioxidant gene responses, by sequestering its inhibitory protein, Kelch-like ECH-associated protein 1 (Keap1). This allows Nrf2 to translocate into the nucleus and bind to the antioxidant response elements, upregulating genes such as NAD(P)H quinone oxidoreductase 1 and heme oxygenase-1 [74,91]. These results reveal that DJ-1 not only directly interacts with the mitochondria to regulate the oxidative stress but also activates the responsible transcription factors to express more antioxidant enzymes to further protect the cells.

DJ-1 modulates dopamine homeostasis in dopaminergic neurons via both transcriptional and post-translational mechanisms. At the transcriptional level, DJ-1 promotes expression of human tyrosine hydroxylase enzymes by inhibiting SUMOylation of the co-repressor pyrimidine tract-binding protein-associated splicing factor (PSF), thereby enhancing transcriptional activity [92]. Post-translationally, DJ-1 enhances dopamine handling by upregulating vesicular monoamine transporter 2 (VMAT2), facilitating dopamine uptake into synaptic vesicles [93], and by binding to the dopamine transporter (DAT), thereby increasing its surface localization and promoting reuptake from the synaptic cleft (Figure 3) [94].

Ferroptosis is defined as ‘an iron-dependent non-apoptotic cell death’ [95]. The phospholipids in membranes can react with free Fe^2+^ or ROS to form phospholipid peroxides (PLOO^−^). Accumulation of PLOO^−^ is the main driver of ferroptosis [96]. Under conditions of cysteine deficiency, DJ-1 promotes the transsulfuration pathway by enhancing S-adenosyl homocysteine hydrolase (SAHH) enzyme activity, enabling cysteine synthesis from methionine [88]. Cysteine is required to generate glutathione (GSH), which is a tripeptide antioxidant acting as a cofactor for glutathione peroxidase enzymes to detoxify lipid peroxides and prevent ferroptosis occurrence (Figure 3) [96].

Finally, DJ-1 has been debated as to whether it acts as a deglycase or glyoxylase, protecting neurons against toxicity induced by glycated α-syn (Figure 3) [97]. This role of DJ-1 is extensively discussed in the following section. The diverse cellular roles of DJ-1 are summarized in Figure 3.

### 3.3. Interaction Between DJ-1 and α-Synuclein

The initial evidence emerged in the mid-2000s, suggesting a potential interaction between DJ-1 and α-syn. In 2004, Shendelman and colleagues initially demonstrated that DJ-1 binds to α-syn both in vitro and in vivo [98]. In vitro, DJ-1 inhibits the formation of α-syn aggregates at early stages before the generation of the mature fibrils. Using DJ-1 homozygous knockout cells showed that α-syn tends to accumulate and result in the formation of insoluble fibrils in the absence of DJ-1 [98]. Following that, Meulener et al. (2005) reported using brain homogenates from healthy, Pick’s disease, or multiple system atrophy patients and found that almost 3% of α-syn was found to physically interact with DJ-1 in all samples, suggesting the DJ-1-α-syn interaction in both physiological and pathological conditions [99]. Further studies strengthened these observations. In N27 dopaminergic cells, overexpression of WT DJ-1 inhibited the formation of A53T α-syn aggregates by 60% [100], which previously had been shown to have a high tendency to aggregate, and these aggregations are cytotoxic and cause cell death [101]. Heat Shock Protein 70 upregulation was shown to be the reason for low α-syn aggregations [100]. Importantly, it was stated that the C106 residue in DJ-1 must be partially oxidized to reach its sulfinic acid form to be effectively able to inhibit α-syn aggregation, as the quiescent unoxidized form of DJ-1 failed to inhibit the aggregations. This indicates that oxidized DJ-1 acts as a chaperone to prevent α-syn aggregation [102].

Later, Zondler et al. (2014) confirmed the aforementioned findings using H4 cells, demonstrating that DJ-1 notably reduced α-syn oligomerization and toxicity through direct interactions. They further showed that pathogenic DJ-1 mutations (L166P, M26I, and L10P) impaired the DJ-1–α-syn interaction by almost 86% [103]. More recently, partially oxidized DJ-1 was found to inhibit the primary nucleation phase of α-syn aggregation directly, hence precluding protofibril formation and further fibril elongation (Figure 4A). Interestingly, in differentiated SH-SY5Y cells, partially oxidized DJ-1 could remodel mature fibrils into oligomeric forms, some of which were cytotoxic, indicating a dual effect [104]. While earlier work proposed that the DJ-1–α-syn interaction may be indirect via docking proteins such as clathrin [105], accumulating evidence now supports direct binding of both quiescent [103] and oxidized DJ-1 [102,104] to α-syn monomers and oligomers.

DJ-1 also modulates α-syn degradation via CMA. Xu et al. (2017) illustrated using SH-SY5Y cells that DJ-1 stabilizes LAMP-2A receptors, enabling efficient degradation of α-syn via the Hsc70–LAMP-2A pathway. Loss of DJ-1 destabilized LAMP-2A, impairing CMA activity and leading to α-syn aggregation (Figure 4C) [106]. Thus, DJ-1 protects against α-syn toxicity both directly by binding and inhibiting aggregation and indirectly by promoting lysosomal clearance.

DJ-1 has been reported to counteract α-syn glycation (Figure 4B). However, the mechanism is still controversial. The controversy has been between whether DJ-1 acts as a glyoxalase enzyme detoxifying a free glycating agent (e.g., methylglyoxal) [107,108,109,110,111,112] or a deglycase removing the covalent bonds [97,113,114]. Lee et al. (2012) reported that DJ-1 acts as a GSH-independent glyoxalase, and DJ-1 treatment increased the viability of SH-SY5Y cells by lowering the levels of the intracellular glyoxals and glyoxal-induced apoptosis [107]. Mutant DJ-1 (L166P and C106A) was unable to protect the cells [107]. By contrast, Richarme et al. (2015) proposed that DJ-1 is a deglycase, repairing proteins by removing the MGO-induced early glycation adducts (e.g., aminocarbinols and hemithioacetal) and releasing the deglycated form of the protein together with D-lactic acid. Hence, DJ-1 prevents the progression of early glycation products into AGEs [113]. DJ-1 failed to deglycate Schiff bases, suggesting that its deglycase activity is selective for certain early adducts but not the very first condensation products of glycation [113]. More recent reports question the deglycase model, showing that DJ-1 is unable to remove glycation of early or advanced glycation endpoints [112]. Gao et al. (2023) showed that co-incubation of DJ-1 and free MGO resulted in a decrease in the MGO levels and the release of L-lactate, hence attenuating protein glycation [112]. In addition, Mazza et al. (2022) also validated the glyoxalase activity of DJ-1, albeit as a weak glyoxalase in vitro [109]. Comparing DJ-1 and glyoxalase 1 showed that DJ-1 is a weaker detoxifier of MGO and DJ-1 has comparatively a minor role in protecting against MGO-induced toxicity. A 2025 study by Mathas et al. demonstrated, using a novel fluorescence-based liquid chromatography assay, that recombinant DJ-1 exhibits both glyoxalase and deglycase activity, with glyoxalase activity predominating [115]. Overall, these findings suggest that DJ-1 safeguards neurons from α-syn toxicity through multiple mechanisms: directly binding and preventing aggregation, facilitating CMA-dependent clearance, and mitigating glycation. Loss of DJ-1 function is therefore predicted to elevate intracellular α-syn, driving aggregation, toxicity, and ultimately neurodegeneration. The potential interactions between DJ-1 and α-syn are summarized in Figure 4.

## 4. Therapeutic Implications

DJ-1 is implicated in the familial and sporadic PD pathogenesis as per the aforementioned reports [66]. This dual involvement makes DJ-1 a promising therapeutic target to treat both PD forms. One potential strategy is the application of small molecules, which stabilize or enhance the functional (oxidized or reduced) forms of DJ-1, which may offer neuroprotection against α-syn aggregation and suppress the loss of dopaminergic neurons. Microinjection of DJ-1 into the SN of PD-model rats mitigated the tyrosine hydroxylase reduction and the death of dopaminergic neurons in the striatum and SN, respectively, while also inhibiting the expression of α-syn [116]. Another approach involves preventing the formation of the inactive sulfonic form of DJ-1. ‘Compound B’ was shown to bind the C106 residue and inhibit its extensive oxidation, thereby preserving DJ-1 activity [117]. Small-molecule activators such as UCP0045037, UCP0054278, and compound-23 have also been identified via in silico screens and shown to enhance DJ-1’s neuroprotective functions against oxidative stress in animal models of PD and ischemic injury [118]. In addition, a novel biologic agent, RNS60, has been demonstrated to upregulate DJ-1 in dopaminergic neurons via the CREB-CBP axis, offering a pharmacological route to enhance endogenous DJ-1 levels [119].

Other than preclinical evidence, patient-based studies support the clinical relevance of DJ-1. Mutations in *PARK7* were originally identified in familial PD cohorts [75], and subsequent reports have confirmed DJ-1 variants (M26I, D149A, A104T, E64D, P158Δ) in other patient populations [76,77,78,79]. Furthermore, elevated levels of oxidized DJ-1 have been detected in erythrocytes and cerebrospinal fluid of sporadic PD patients compared with healthy controls, suggesting that DJ-1 may serve as a potential biomarker for diagnosis and disease monitoring [120,121]. Specifically, one study detected high levels of C106-oxidized DJ-1 in erythrocytes of unmedicated PD patients relative to medicated patients and healthy control subjects using specific antibodies against C106-oxidized DJ-1 [122].

Altogether, these findings highlight DJ-1 as both a potential therapeutic target and a candidate biomarker. However, further clinical research will be required to establish whether DJ-1 alterations can be consistently validated in large patient cohorts and whether modulation of DJ-1 function can translate into meaningful clinical benefit.

## 5. Conclusions

Despite extensive research, the precise mechanisms by which DJ-1 protects against pathological α-syn in PD are yet to be understood. Nevertheless, collective evidence clearly highlights DJ-1 as a multifunctional protein primarily involved in protecting neurons from various insults, including oxidative stress, mitochondrial dysfunction, ferroptosis, and α-syn–mediated toxicity. A clearer understanding of how DJ-1 interacts with α-syn is of particular importance. Current data strongly suggest that this interaction is essential for maintaining α-syn at low, non-aggregating levels and thereby promoting neuronal survival. Clinical studies have also reported altered DJ-1 levels and isoforms in the blood of PD patients, supporting its potential as a biomarker [118,119,120]. Future studies should further dissect the molecular details of DJ-1–α-syn crosstalk and clarify its clinical significance in patient cohorts, which will be essential for translating mechanistic insights into therapeutic strategies for both familial and sporadic PD.

## Figures and Tables

**Figure 2 brainsci-15-00899-f002:**
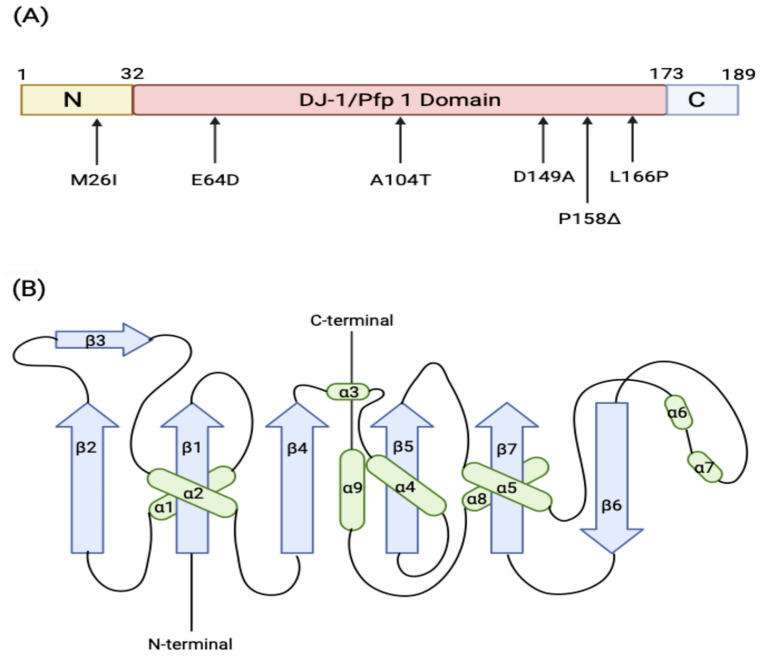
**Structure of DJ-1 monomer.** (**A**) Three DJ-1 domains and familial mutations of DJ-1 associated with PD. (**B**) DJ-1 monomer topology. The α-helices are represented by ovals (green) and the β-strands (purple) by arrows (Created in Biorender.com) [27].

**Figure 3 brainsci-15-00899-f003:**
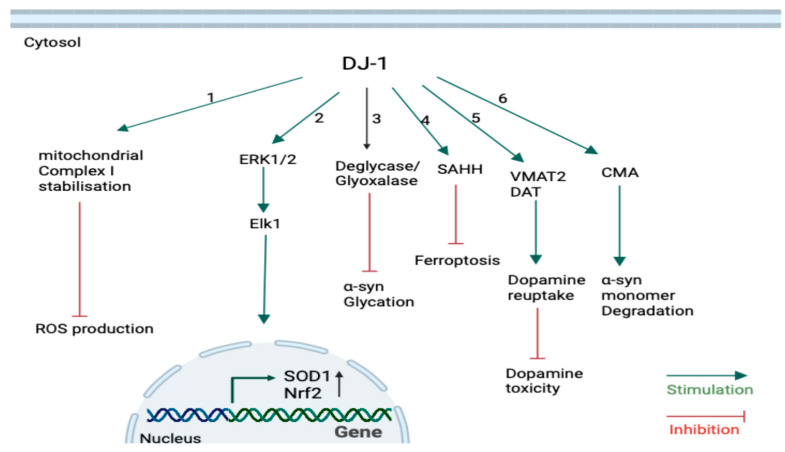
**Intracellular roles of DJ-1.** (1) Under oxidative stress, DJ-1 is translocated into the mitochondria to stabilize complex I activity to prevent ROS production. (2) DJ-1 can also reduce ROS production by indirectly upregulating SOD1 and Nrf2 genes via the ERK1/2-Elk1 pathway. (3) DJ-1 protects neurons against toxicity induced by glycated α-syn by acting either as a deglycase or glyoxalase or both. (4) DJ-1 promotes the activity of the enzyme SAHH, leading to detoxifying lipid peroxides and inhibiting ferroptosis. (5) DJ-1 modulates neuronal dopamine homeostasis. It facilitates dopamine reuptake by increasing the expression of VMAT2 and localization of DAT on the terminal ends, thereby inhibiting dopamine-induced toxicity. (6) α-syn monomers are efficiently degraded in the presence of DJ-1 by CMA, preventing α-syn aggregation and toxicity (created in Biorender.com) [27].

**Figure 4 brainsci-15-00899-f004:**
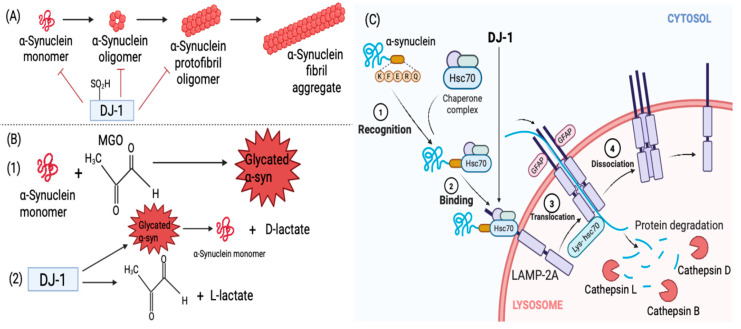
**Potential direct and indirect interaction of DJ-1 with α-syn.** (**A**,**B**) are direct mechanisms by which DJ-1 inhibits α-syn. (**A**) The sulfinated form (fully active) of DJ-1 directly interacts with α-syn monomers, oligomers, and protofibrils, preventing the formation of α-syn fibrils and their maturation. (**B**) (1) Excess production of intracellular glycating agents such as MGO can lead to the glycation of α-syn. (2) DJ-1 has been proposed to counter this either by acting as a deglycase, removing early glycation adducts, regenerating native α-syn, and producing D-lactate, or, as a glyoxalase, detoxifying MGO to L-lactate, thereby preventing glycation, or potentially through both activities. (**C**) Indirect mechanism of DJ-1 via CMA. DJ-1 promotes CMA activity to facilitate the degradation of soluble α-syn monomer. In the absence of DJ-1, CMA activity is suppressed, leading to accumulation and aggregation of α-syn. ① Recognition: Hsc70 chaperone protein recognizes the KKFERQ-like motif on α-syn. ② Binding: Hsc70 binds to α-syn, forming the Hsc70/α-syn complex. In the presence of DJ-1, the Hsc70/α-syn complex is successfully associated with LAMP-2A monomers on the lysosomal membrane. Upon their association, LAM-2A is multimerized to generate a translocation complex by the assistance of GFAP. ③ Translocation: α-syn is unfolded and translocated into the lysosomal lumen, where the degradation proteins such as Cathepsin B, D, and L break down α-syn. ④ Dissociation: LAMP-2A multimers are disassembled into inactive monomers (created in BioRender.com) [27].

## Data Availability

No new data were created or analyzed in this study.

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
