# Peer review of "Interaction Between α-Synuclein and DJ-1 in Parkinson’s Disease"

_brainsci, 2025, doi:10.3390/brainsci15090899_

Round 1
Reviewer 1 Report
Comments and Suggestions for Authors
I commend the authors on exploring the interaction a-synuclein and DJ-1 in PD. They conclude that the precise mechanisms by which DJ-1, is yet to be understood. Evidence indicates that DJ-1 is a multifunctional protein primarily involved in protecting neurons from various insults.
The topic is clinically meaningful, and the effort to synthesize evidence is appreciated. Your work addresses a crucial question in the field of movement disorders. To enhance clarity, methodological transparency, and interpretability of results, we respectfully suggest the following improvements, structured according to manuscript sections.
SECTION-BY-SECTION COMMENTS
1. Title and Abstract
Comment: The title accurately reflects the study objective and design. The abstract is generally well structured.
2. Introduction
Comment: The background and the rationale is clearly articulated, but i suggest to add next to ref 5 this ref. (Whole body-electromyostimulation effects on serum biomarkers, physical performances and fatigue in Parkinson’s patients: A randomized controlled trial, by DI cagno et al 2023) to further contextualize the clinical impact of motor symptoms on PD
2.1 Structure
Phrase "Recombinant a-syn....at very low concentrations" is not clear and should be rewrittren
I feel that coclusions are exhaustive
Table are well represented.
Figures: i suggest to improve the quality of figure 2 A.
Author Response
You can find my response in the attachment.

Reviewer 2 Report
Comments and Suggestions for Authors
The review presents a well-supported hypothesis regarding the interaction between DJ-1 and α-synuclein in Parkinson's disease (PD).
The manuscript can be improved in several areas regarding clarity, academic polish, and better communication of scientific ideas.
- Some sentences are excessively long or dense with multiple concepts.
E.g: Page 1: “The precise function of α-syn in the healthy human brain is not agreed but it has been reported to play a role in vesicular trafficking…” → consider splitting for clarity.
Page 10: "Reports also mentioned that DJ-1 provides this protection against toxicity induced by α-syn aggregation by a direct interaction." → slightly redundant.
Thus, Reduce the length or awkwardness of long sentences by simplifying them. Whenever possible, use the active voice.
- Although the hypothesis is strong, it is not always explicitly supported. Reiterate the central hypothesis in the abstract and discussion - DJ-1 regulates synuclein levels and aggregation. Provide an explanation of how this may bridge familial and sporadic PD mechanisms.
- Though figures are well-made, the narrative doesn't always integrate them deeply.
On page 3 of Figure 1, explain briefly how the mutations shown (A53T, E46K) may lead to aggregation susceptibility.
Figure 4 on page 12: clarify how the DJ-1-induced remodeling of mature fibrils into cytotoxic oligomers could paradoxically increase toxicity - this seems to contradict the overall neuroprotective narrative.
There are several concepts that are repeated excessively, including DJ-1 oxidation states and α-syn aggregation types.
- The discussion on tetramer vs. monomer appears twice. Consolidate discussions where concepts are repeated.
- Provide a summary of the current consensus or suggest future experiments (e.g., new assays or in vivo validations) to resolve such controversies.
- Rephrase Page 1, Abstract: "This review hypothesizes that..." for consistency with academic style.On page 4, "WT" (wild-type) needs to be defined.
- A table summarizing key experimental findings:Eg, direct vs. indirect DJ-1 effects, type of model used (eg, cell, mouse), key outcomes.
- You've reserved this for Section 4. Briefly mention it in the Introduction or Abstract to motivate the relevance.
- Could this DJ-1/α-syn interaction model be extended to other synucleinopathies (e.g., MSA or DLB)? The discussion could include a brief mention of this.
- Maintain consistency in citation formatting (some include initials, some don't). Depending on the journal, you should check this before submitting.
Author Response

(The authors gave the same response as above.)

Reviewer 3 Report
Comments and Suggestions for Authors
This review manuscript on Parkinson’s disease (PD) explores the link between DJ-1 and α-synuclein, aiming to highlight the potential importance of DJ-1 in PD research. The authors begin with a background on the well-established role of α-synuclein in PD and subsequently delve into the molecular and cellular functions of DJ-1, with an attempt to suggest its translational relevance.
While the manuscript provides some structural and pathological insights into DJ-1, its clinical and translational significance in PD remains underdeveloped. The authors fall short in effectively situating DJ-1 within the broader landscape of PD etiopathogenesis. Specifically, the manuscript does not sufficiently connect DJ-1 to key pathogenic mechanisms such as α-synuclein aggregation, mitochondrial dysfunction, oxidative stress, or impaired protein clearance. As a result, the narrative lacks depth and fails to deliver a compelling rationale for the importance of DJ-1 in the context of PD research.
Furthermore, the graphical illustrations are overly simplistic and do not adequately capture the complexity of PD as a multifactorial neurodegenerative disease.
Lastly, PD is a human disease. Clinical reference and the findings from medicine journals have to be included.
A major revision is required to restructure the manuscript, strengthen the mechanistic connections, and present a more integrated and impactful view of DJ-1’s role in PD pathophysiology, in order to make the manuscript to be a better story.
Author Response

(The authors gave the same response as above.)

Round 2
Reviewer 2 Report
Comments and Suggestions for Authors
The authors have addressed the suggested changes. The manuscript may be accepted after English editing if required.
Reviewer 3 Report
Comments and Suggestions for Authors
n/a